# Urine of Cats with Severe Fever with Thrombocytopenia Syndrome: A Potential Source of Infection Transmission

**DOI:** 10.3390/pathogens14030254

**Published:** 2025-03-05

**Authors:** Hirohisa Mekata, Mari Yamamoto, Yasuyuki Kaneko, Kentaro Yamada, Tamaki Okabayashi, Akatsuki Saito

**Affiliations:** 1Center for Animal Disease Control, University of Miyazaki, 1-1 Gakuen-Kibanadai-Nishi, Miyazaki 889-2192, Japan; markoba@cc.miyazaki-u.ac.jp (M.Y.); kentaro-y@cc.miyazaki-u.ac.jp (K.Y.); okbys81@cc.miyazaki-u.ac.jp (T.O.); sakatsuki@cc.miyazaki-u.ac.jp (A.S.); 2Veterinary Teaching Hospital, Faculty of Agriculture, University of Miyazaki, 1-1 Gakuen-Kibanadai-Nishi, Miyazaki 889-2192, Japan; yasuyuki-kaneko@cc.miyazaki-u.ac.jp; 3Department of Veterinary Sciences, Faculty of Agriculture, University of Miyazaki, 1-1 Gakuen-Kibanadai-Nishi, Miyazaki 889-2192, Japan; 4Graduate School of Medicine and Veterinary Medicine, University of Miyazaki, 5200 Kiyotakecho Kihara, Miyazaki 889-1692, Japan

**Keywords:** cats, DC-specific ICAM-3-grabbing nonintegrin, personal protective equipment, SFTS phlebovirus, Vero cells

## Abstract

Severe fever with thrombocytopenia syndrome (SFTS), caused by infection with the SFTS virus, is an emerging fatal tick-borne zoonosis endemic to East Asia. Although SFTS is a tick-borne disease, the virus can be transmitted from animals with SFTS without a tick bite. Direct transmission of the SFTS virus from animals to humans has been reported; however, the transmission route is unclear in some cases. Therefore, this study focused on the possibility of SFTS virus transmission through urine and attempted to isolate the infectious virus from the urine of animals with SFTS. Since more efficient cell isolation is needed to determine whether the SFTS virus is present, we first expressed dendritic cell-specific ICAM-3-grabbing nonintegrin (DC-SIGN), the major receptor for the virus, in Vero cells (Vero-DC-SIGN cells) using a retroviral vector. When inoculated with equal amounts of the SFTS virus strain and SFTS-virus-infected animal serum, Vero-DC-SIGN cells had 42–136% and 20–85% more foci, respectively, than their parent Vero cells. After confirming that Vero-DC-SIGN cells were more suitable for the isolation of the SFTS virus, we investigated whether it could be isolated from the urine of eight cats and two dogs with SFTS. The virus was isolated from 25 μL of urine from two cats with SFTS. Considering that cats excrete 50–100 mL of urine per day, the transmission of the SFTS virus via the urine of cats with SFTS cannot be ruled out. Individuals examining or caring for cats suspected of having SFTS should be aware of the possibility of viral transmission via urine.

## 1. Introduction

Severe fever with thrombocytopenia syndrome (SFTS) is an emerging tick-borne viral zoonosis in East Asia that is fatal in 10–27% of patients [1,2,3]. The SFTS virus (SFTSV) is the causative agent of SFTS and is classified as a species of *Bandavirus dabieense* in the genus *Bandavirus* of the family *Phenuiviridae* (International Committee on Taxonomy of Viruses, https://ictv.global/report/chapter/phenuiviridae/phenuiviridae/bandavirus, accessed on 12 February 2025). SFTSV was first isolated from a patient with an acute febrile illness in China in 2009 [4]. To date, the endemicity of SFTSV has been confirmed in China, South Korea, and Japan [5,6,7]. Although it is unclear whether SFTS is as endemic as in the three countries mentioned above, cases of SFTS have been confirmed in other Asian countries [8,9,10,11,12]. SFTSV infects a wide range of animal species in nature [13]. However, most animals are considered asymptomatic when infected with SFTSV, and only a limited number of species develop severe viremia [14]. Domestic cats are more likely to develop severe symptoms of SFTSV infection than other animals, and the fatality rate is approximately 60% [15,16]. Domestic dogs are often asymptomatic when infected with SFTSV [17,18]. Although the factors that contribute to the severity of SFTS in dogs are unclear, some dogs naturally infected with SFTSV develop severe symptoms [19,20]. SFTSV is primarily transmitted by tick bites, with *Haemaphysalis longicornis* and *H. flava* thought to be the primary vectors [21,22]. However, SFTSV can be transmitted directly from an SFTSV-infected animal to humans without a tick bite, and close contact with infected animals is a risk factor for SFTSV infection in humans [1,23,24].

Recently, the direct non-tick transmission of SFTSV from infected animals to veterinary personnel or pet owners has become a significant public health concern in Japan [1,25,26,27,28,29,30]. The seroprevalence of SFTSV among small animal veterinarians and nurses in SFTS-endemic areas of Japan was reported to be 2.2% (2/90) in Miyazaki and 4.2% (3/71) in Nagasaki, which is higher than that of healthy blood donors in Miyazaki (0%, 0/1000) and people over 50 years of age in Ehime (0.14%, 1/692) [31,32,33]. Some reports of direct animal-to-human transmission have indicated that bites from or direct contact with the blood of animals with SFTS are the source of viral transmission [1,26,30]. In contrast, there have been cases in which veterinary personnel had only brief contact with cats infected with SFTSV, despite wearing masks and gloves and not receiving external injuries from animals with SFTS [25,28,29]. Recently, SFTSV was isolated from the urine of two SFTSV-infected dogs in Toyama, Japan [34]. However, it is unclear whether infectious viral particles are shed in the urine of cats, where the incidence of SFTS is much higher than that in dogs. Therefore, it is important to confirm whether the infectious SFTSV is shed in the urine of cats with SFTS.

Cells used for virus isolation must be efficiently infected with the target virus and able to efficiently propagate it. Vero cells are the most commonly used cell line for SFTSV isolation, as they have been used in reports on the first isolation of SFTSV from patients in China, South Korea, and Japan [4,35,36]. However, Vero cells do not express dendritic cell-specific ICAM-3-grabbing nonintegrin (DC-SIGN), a major receptor of SFTSV [37]. Therefore, the gene transfer of DC-SIGN into Vero cells may make them more suitable for SFTSV isolation.

In this study, we aimed to generate Vero cells that stably expressed DC-SIGN (Vero-DC-SIGN cells) and confirm that they were more susceptible to SFTSV infection than their parental Vero cells. Subsequently, we attempted to isolate SFTSV from the urine of cats and dogs with SFTS using Vero-DC-SIGN cells. This study contributes to both animal and public health by controlling the spread of SFTS.

## 2. Materials and Methods

### 2.1. Animals

Two dogs (male, 4 and 12 years old) and eight cats (6 males and 2 females, 1–6 years old) from Oita and Miyazaki, Japan, were examined at the corresponding veterinary clinic. These animals were naturally infected with SFTSV and exhibited severe health conditions.

### 2.2. Compliance with Ethical Standards and Biosafety

Blood and urine samples were collected by clinical veterinarians to diagnose SFTS according to the ethical guidelines of their respective hospitals. Some animals were monitored to determine the optimal time for hospital discharge and to prevent nosocomial infections. The remaining samples were used for research purposes with the consent of the animal owner. The isolation of infectious SFTSV and all experiments using infectious SFTSV were performed in a Biosafety Level 3 laboratory at the University of Miyazaki, following standard guidelines.

### 2.3. Generation of Vero-DC-SIGN Cells

To generate Vero-DC-SIGN cells, retroviral vectors were used to transduce human DC-SIGN cDNA. First, the cDNA encoding human DC-SIGN was ligated into the pDON-5 Neo DNA plasmid (TaKaRa Bio, Kusatsu, Japan), which carries the neomycin resistance gene, using an In-Fusion cloning method. The cDNA was derived from pcDNA3-DC-SIGN obtained from the AIDS Research and Reference Reagent Program of the National Institute of Allergy and Infectious Disease, National Institutes of Health [38]. The generated plasmids were amplified in *Escherichia coli* DH5α (TaKaRa Bio). After cloning and culturing *E. coli*, the plasmids were extracted from it using NucleoSpin Plasmid EasyPure (TaKaRa Bio). The plasmid, confirmed by Sanger sequencing to have no unintended mutations in the DC-SIGN cDNA, was designated as pDON-5 Neo DC-SIGN. To generate a recombinant retrovirus for gene transfection, pDON-5 Neo DC-SIGN was co-transfected with a packaging vector (pGP vector, TaKaRa Bio) and pMD.2G plasmid into Lenti-X 293T cells (TaKaRa Bio) using the Xfect Transfection Reagent (TaKaRa Bio). The pMD2.G was a gift from Dr. Didier Trono (cat. # 12259; https://www.addgene.org/12259/, accessed on 12 February 2025). Recombinant virus particles were harvested from cell supernatants 48 h after transfection. After the virus was concentrated with a polyethylene glycol solution, the recombinant retrovirus was inoculated into Vero cells for transduction with DC-SIGN. Forty-eight hours after inoculation, the culture medium was replaced with a medium containing G418 (final concentration: 1600 μg/mL) to select transgenic Vero cells. The transgenic Vero cells were cloned using the limiting dilution method.

### 2.4. Confirmation of DC-SIGN Expression in Vero-DC-SIGN Cells

The expression of DC-SIGN in the transduced Vero cells was confirmed by Western blotting. Briefly, cell lysates were denatured in 4× Bolt LDS Sample Buffer (Thermo Fisher Scientific, Waltham, MA, USA) containing 2% 2-mercaptoethanol at 70 °C for 10 min. Samples were loaded onto SDS-PAGE, and proteins were transferred to PVDF membranes with dry transfer using an iBlot 2 Device (Thermo Fisher Scientific). The PVDF membranes were blocked with Bullet Blocking One for Western blotting (Nacalai, Kyoto, Japan) and probed with mouse anti-DC-SIGN monoclonal antibody (Abcam, Cambridge, UK) diluted 2500-fold with Signal Enhancer HIKARI for Western blotting and ELISA (Nacalai) for 1 h at 25 °C with shaking. After extensive washing in water and tris-buffered saline with 0.5% Tween 20, the membranes were probed with HRP-conjugated goat anti-mouse polyclonal IgG antibodies (LGC Clinical Diagnostics, Milford, MA, USA) diluted 2500-fold for 1 h at 25 °C with shaking. Probed proteins were visualized using Western BLoT Ultra-Sensitive HRP Substrate (TaKaRa) and imaged using a ChemiDoc Touch system (Bio-Rad Laboratories, Hercules, CA, USA). After the probes were stripped from the PVDF membrane using WB Stripping Solution Strong (Nacalai), β-actin was probed with peroxidase-labeled anti-mouse β-actin monoclonal antibody (MilliporeSigma, Burlington, MA, USA).

### 2.5. SFTSV Strains and Serum of SFTSV-Infected Animals

SFTSV strain A17 (Accession No. LC536536, 46, 56) was isolated from a tick in Miyazaki, Japan, in 2016 and was classified as the most common genotype J1 (also called B-2) in Japan [39]. SFTSV strain SG403 was isolated from the serum of a cat infected with SFTSV in Saga, Japan, in 2023. The partial nucleotide sequence of the S genome of SFTSV was determined according to previously described methods [36], and its genotype was C4 (also called A), which is genetically related to strains prevalent mainly in China. The A17 strain was isolated from Vero cells, and the SG403 strain was isolated from Vero-DC-SIGN cells. The supernatant from the three culture passages was stored at −80 °C until used for analysis.

Sample numbers 230413-3-DOG-SE and 231127-4-CAT-SE are serum samples collected from a dog with SFTS in April 2023 and a cat with SFTS in November 2023, respectively. Clinical veterinarians who examined each animal suspected that the animal had SFTS and requested laboratory tests from the Center for Animal Disease Control, University of Miyazaki. The day the owner noticed the animal’s fever or other physical symptoms was defined as day 0 of the onset of the disease. As a result of a quantitative real-time reverse transcription polymerase chain reaction (qRT-PCR), these animals were found to be infected with SFTSV. The remaining serum samples were stored at −80 °C until used for analysis.

### 2.6. RT-PCR

Our laboratory accepts testing for SFTSV infection in dogs and cats using qRT-PCR at the request of clinical veterinarians, which was performed as follows: RNA was extracted from serum or urine samples collected by clinical veterinarians using a fully automated nucleic acid extraction system (MagLEAD System; Precision System Science, Chiba, Japan). The SFTS viral genome was detected using One-Step PrimeScript III RT-qPCR Mix (TaKaRa Bio), target gene-specific primers, and FAM probes (PrimeTime qPCR assays, Integrated DNA Technologies, Coralville, IA, USA) on a LightCycler 96 system (Roche, Basel, Switzerland). GAPDH mRNA was simultaneously detected using the TEX615 probe as an internal standard. Primer and probe sets used to detect SFTSV and GAPDH mRNA were adapted from previous reports [40,41]. Some of the reverse primer sequences for GAPDH mRNA were corrected for duplicate bases because the sequences did not match those of animals other than cats. SFTSV forward primer, TGTCAGAGTGGTCCAGGATT; reverse primer, ACCTGTCTCCTTCAGCTTCT; probe, FAM-TGGAGTTTG/ZEN/GTGAGCAGCAGC-Iowa Black Dark Quencher. GAPDH forward primer, GCTGCCCAGAACATCATCC; reverse primer, GTCAGATCCACRACBRAYAC; and probe, TEX615-TCACTGGCATGGCCTTCCGT-Iowa Black Dark Quencher.

### 2.7. Comparison of SFTSV Infection Efficiency Using Focus-Forming Assay

The SFTS virus strains A17 and SG403 were diluted to a concentration of approximately 5 × 10^2^ focus-forming units (FFUs)/mL in parental Vero cells. Serum from SFTSV-infected animals (230413-3-DOG-SE and 231127-4-CAT-SE) was diluted 10^2^–10^4^ times. Dilutions of 200 μL were applied to a monolayer of Vero-DC-SIGN cells or their parental Vero cells in a 24-well plate and incubated at 37 °C for 1 h. After sample removal and washing with PBS, the cells were overlaid with a maintenance medium containing 1% methylcellulose and incubated in a 5% CO_2_ incubator at 37 °C. After 4 days, the cells were fixed with 4% paraformaldehyde (Nacalai) for 1 h. After two washes with PBS, 200 μL of a 3000-fold diluted mouse anti-SFTSV NP antibody was added to the cells and incubated at 37 °C for 1 h. The mouse anti-SFTSV NP antibody was generated using DNA immunization as described previously [42]. After three washes with PBS, 200 μL of a 2500-fold diluted goat anti-mouse IgG antibody (AlexaFluor488 anti-mouse IgG, Thermo Fisher Scientific) was added and incubated at 37 °C for 1 h. After three washes with PBS, cells were observed for fluorescence using a fluorescence microscope (EVOS M7000 Imaging System, Thermo Fisher Scientific).

### 2.8. Virus Isolation from Urine

Urine samples were centrifuged, filtered through a 0.45 μm filter, and serially diluted 4–16-fold with Opti-MEM (Thermo Fisher Scientific) to adjust to a pH suitable for cell culture. Two hundred microliters of each diluted urine sample was applied to a monolayer of Vero-DC-SIGN cells, and then the focus-forming assay protocol described above was followed. Viral isolation was considered positive if fluorescence was observed in one or more of the four wells.

### 2.9. Statistical Analysis

A Student’s *t*-test was used to compare the number of foci between Vero-DC-SIGN and Vero cells infected with SFTSV. Analyses were performed using GraphPad Prism 6 software (GraphPad Software, San Diego, CA, USA). A *p*-value < 0.05 was considered statistically significant.

## 3. Results

### 3.1. Evaluation of Vero-DC-SIGN

To increase the efficiency of SFTSV isolation, Vero cells stably expressing human DC-SIGN, a major cell entry receptor for SFTSV, were generated. Vero cells were transduced with a retroviral vector expressing human DC-SIGN. After selection with neomycin and cloning using the limiting dilution method, cells that grew well and were confirmed to express DC-SIGN with Western blotting were designated as Vero-DC-SIGN cells (Figure 1). To compare the susceptibility of Vero-DC-SIGN cells and their parental cells to SFTSV, both were infected with equal amounts of the SFTSV strains A17 and SG403, and the number of foci in each cell was measured. In Vero-DC-SIGN cells, the number of foci increased by 42% for SFTSV strain A17 and 136% for strain SG405 compared with that in the parental cells (Figure 2a). Comparing the images 4 days after virus inoculation, clear fluorescence was observed in Vero-DC-SIGN cells compared with that in the parental cells (Figure 2b). We compared the susceptibility of these cells to SFTSV using frozen serum from SFTSV-infected animals, assuming that the virus could be isolated from clinical samples. In Vero-DC-SIGN cells, the number of foci increased by 85% in SFTSV-infected cat serum (sample ID: 231127-4-CAT-SE) and by 20% in SFTSV-infected dog serum (sample ID: 230413-3-DOG-SE) compared with parental cells (Figure 2c). These results suggest that Vero-DC-SIGN cells are more suitable than parental Vero cells for isolating SFTSV from clinical samples.

### 3.2. Isolation of Infectious SFTSV from Cat Urine

To determine whether the urine of SFTSV-infected animals was a source of infection, we attempted to isolate SFTSV from their urine samples. A total of 33 urine samples (eight cats and two dogs) in which SFTS viral RNA was detected and the Cq value was less than 35 were used for virus isolation. When the cells were inoculated with undiluted or twofold diluted urine samples, they detached from the wells. Therefore, the cells were inoculated with four-, eight-, or sixteenfold dilutions of urine samples in a buffered medium. An average of 10 foci were observed in two of the four wells inoculated with a fourfold dilution of 230214-1-CAT-Day2 urine (Figure 3). Foci were also observed in one of the four wells inoculated with eightfold diluted fecal urine. A total of 31 PCR-positive but isolation-negative urine samples showed no foci at all, and the fluorescence of the two foci indicated that they were less likely to be false positives. In 240122-1-CAT-Day10, although the cells inoculated with fourfold diluted urine were detached, two foci were observed in one of the four wells inoculated with eightfold diluted urine. When the infectious SFTSV was isolated from the urine of 230214-1-CAT, the amount of SFTSV RNA in the urine and serum had Cq values of 25.4 and 10.3, respectively. In 240122-1-CAT-Day10, the Cq value was 31.3 in the urine. The Cq value of the serum sample collected on the following day was 35.0. Patient 230214-1-CAT died 3 days after the development of SFTS, and SFTSV was isolated from urine collected one day before death. The 240112-1-CAT died 12 days after the development of SFTS, and SFTSV was isolated from urine collected 2 days before death.

## 4. Discussion

This study showed that the stable expression of DC-SIGN, a receptor for viral entry, in Vero cells increased the efficiency of SFTSV isolation. Using Vero-DC-SIGN cells, infectious SFTSV was successfully isolated from 25 μL urine samples of SFTSV-infected cats. In Japan, the number of human SFTS cases increases annually (National Institute of Infectious Diseases, https://www.niid.go.jp/niid/ja/sfts/3143-sfts.html; Japanese version accessed on 12 February 2025). In addition, secondary infections in individuals who examine or care for animals with SFTS have become a concern. A significant problem is that in some cases, it is not clear how SFTSV is transmitted from an animal with SFTS to veterinarians.

A comparison of different cell lines showed that SFTSV is more likely to propagate in Vero and Huh-7 cells [43]. Therefore, it is reasonable to use Vero cells, which are widely used and easy to handle, to isolate and propagate SFTSV. SFTSV is more likely to grow in Vero cells because these cells do not produce type I interferons [44]. SFTSV is asymptomatic in most immunocompetent mice, whereas it replicates and causes severe symptoms in type I interferon receptor-knockout mice [45,46]. DC-SIGN is the cell entry receptor for many viruses; however, Vero cells that do not express DC-SIGN are often selected for isolation [47,48]. Dengue virus is one such virus, and DC-SIGN expression in Vero cells increases the efficiency of dengue virus infection [49,50]. Another problem with isolating and propagating these viruses in Vero cells is that they are forced to use different entry receptors in vitro, which may induce RNA mutations. Amino acid mutations have been observed in repeated passages of SFTSV using Vero cells [51]. Therefore, the use of Vero-DC-SIGN for SFTSV isolation and passage may reduce the risk of Vero-cell-specific mutations in the SFTSV genome.

When equal amounts of SFTSV were inoculated into the cells, >42% of the A17 strain and >136% of the SG403 strain infected Vero-DC-SIGN cells than the parental Vero cells (Figure 2a). A possible reason for the lower increase in the A17 strain compared with the SG403 strain is that, while the A17 strain was isolated with three passages of parental Vero cells, the SG403 strain was isolated using Vero-DC-SIGN cells. Although C-C motif chemokine receptor 2 (CCR2) expression is restricted to monocytes and macrophages, CCR2 has also been reported to function as an entry receptor for SFTSV [52]. The A17 strain may have been less affected by DC-SIGN expression because it is more selective for viruses entering the cell via other receptors expressed on Vero cells. An increase in the number of foci in Vero-DC-SIGN cells was observed not only when inoculated with SFTSV strains but also when inoculated with serum from SFTSV-infected cats and dogs (Figure 2c). Additionally, more intense fluorescence was detected in Vero-DC-SIGN cells (Figure 2b). These results suggest that Vero-DC-SIGN cells are suitable for viral isolation and propagation. The isolation rate of dengue virus from clinical samples was 30.8% in Vero cells and 46.2% in Vero-DC-SIGN cells [50]. For the Junín virus of the family *Arenaviridae*, the expression of DC-SIGN in Vero cells promoted viral entry and increased the infection rate by nearly twofold [53]. The increase in the isolation efficiency of SFTSV caused by the expression of DC-SIGN was also consistent with these reports.

SFTS viral RNA has been detected in the urine of cats, dogs, and ferrets [15,18,54,55]. However, the detection of viral RNA does not necessarily indicate that it is the source of the infection. It is necessary to isolate the infectious virus to determine whether it is present in urine. The reason for focusing on urine in this study was that it is excreted in larger quantities than other body fluids. Normal cats and dogs produce 10–20 and 20–100 mL/kg/day of urine, respectively [56]. Although a cat with SFTS may have reduced urine output, a 5 kg cat will urinate 50–100 mL/day. In this study, 2–10 FFUs of infectious SFTSV were detected in 25 μL of urine. This means that 4 × 10^3^–4 × 10^4^ FFUs/day of infectious SFTSV would be discharged. Therefore, the urine from SFTSV-infected cats may be a direct source of infection in humans and animals. Veterinary personnel have been infected with SFTSV despite not being exposed to external injuries or blood from cats with SFTS [25,28]. It is possible that SFTSV in the urine of cats may invade exposed mucous membranes, such as the eyes. Veterinarians should wear personal protective equipment such as masks, goggles, and gloves when examining cats suspected of having SFTS. Veterinarians should also instruct those caring for cats with SFTS to wear personal protective equipment when handling materials that could be contaminated with cat urine. Ventilation should also be provided as the possibility of airborne transmission of SFTSV cannot be excluded [57,58].

RT-PCR and ELISA performed in our laboratory confirmed simultaneous SFTSV infections in five pairs of cohabiting cats. It is possible that these cohabiting cats were simultaneously infected with SFTSV from different ticks at almost the same time. However, we cannot exclude the possibility that SFTSV was directly transmitted from a cat with SFTS to a cohabiting cat. Direct non-tick transmission from humans infected with SFTSV to other humans has been reported [59,60,61,62]. Because cats are more susceptible to SFTSV than humans, it may be challenging to consider the absence of cat-to-cat SFTSV transmission. There was no significant difference between repellent use and the rate of SFTSV infection in cats [16]. If a certain percentage of direct non-tick transmission from cats with SFTS to cats is included, the lack of a significant difference between the use of repellents and the infection rate is understandable. Controlling SFTSV infection in cats is important not only for animal health but also for public health because cat-to-cat transmission increases the prevalence of SFTSV in ticks living in the human environment, in addition to the risk of direct cat-to-human transmission.

In this study, urine with Cq values ≤ 35 for SFTS viral RNA was used for isolation. Infectious SFTSV was isolated from 2 of the 33 urine samples. It was initially thought that the efficiency of virus isolation from urine would correlate with the amount of SFTS viral RNA. However, it was unexpected that the virus would be isolated from a urine sample of the same cat with a low level of viral RNA (Figure 3). Notably, the two cats died within 2 days of isolation of the infectious virus from their urine. Under normal conditions, infectious SFTSV may not be excreted in the urine; however, if renal function is severely impaired due to SFTSV infection, SFTSV-infected lymphocytes may leak into the urinary system and SFTSV may be present in the urine. Since a pathological autopsy was not performed, it is unclear why 240112-1-CAT died despite being in a recovery period. Saga et al. isolated infectious SFTSV from the urine of two dogs more than 2 weeks after recovery from SFTS [34]. They also reported that SFTS viral RNA was detected in the urine of these dogs 2 months after recovery, although SFTSV was not isolated. In this study, viral RNA was detected in the urine of two dogs on days 55 and 69 of illness (Figure 3). SFTS viral RNA was not detected in the urine of cats recovering from SFTS for more than 1 month. Therefore, the long-term detection of SFTS viral RNA in the urine after recovery may be a specific phenomenon in SFTSV-infected dogs. Although dogs are considered to have a mild disease when infected with SFTSV, more detailed studies of the damage it causes to the canine urinary system are needed. Monitoring the extent of renal damage in SFTSV-infected animals—for example, measuring creatinine and blood urea nitrogen—might help elucidate the pathogenesis of SFTS in dogs and cats. In this study, SFTS viral RNA was detected in 31 of the 33 urine samples; however, no infectious viruses were isolated. It is not known whether this was due to the absence of any infectious SFTSV immediately after urination.

In SFTSV-endemic areas, cats must be kept indoors to avoid ticks and contact with animals infected with SFTS. However, this problem is not easily solved because some cat owners prefer to let their cats roam free, cats sometimes escape from their homes, and there are a certain number of community cats and stray cats whose owners are not clear. Awareness of SFTS among citizens and cat owners is important for controlling SFTS in pets and humans. This study demonstrated the potential for owners and veterinarians to become infected with SFTSV through exposure to urine from cats with SFTS. However, it remains unclear whether direct infection from the urine of cats with SFTS actually occurs and how many FFUs of the virus are required to establish infection in healthy individuals. Further research is required to elucidate these issues. It is important for public health to elucidate the possible routes of viral transmission and take measures to reduce the risk of infection.

## 5. Conclusions

In this study, we generated cells with high isolation efficiency of SFTSV and isolated infectious SFTSV from 25 μL urine samples from two cats with SFTS. The results indicate the possibility of direct transmission from the urine of cats with SFTS. Some veterinarians have been infected with SFTSV despite not being exposed to blood or external injuries from the cats with SFTS they examined. This study represents one possible route of the non-tick-mediated transmission of SFTSV.

## Figures and Tables

**Figure 1 pathogens-14-00254-f001:**
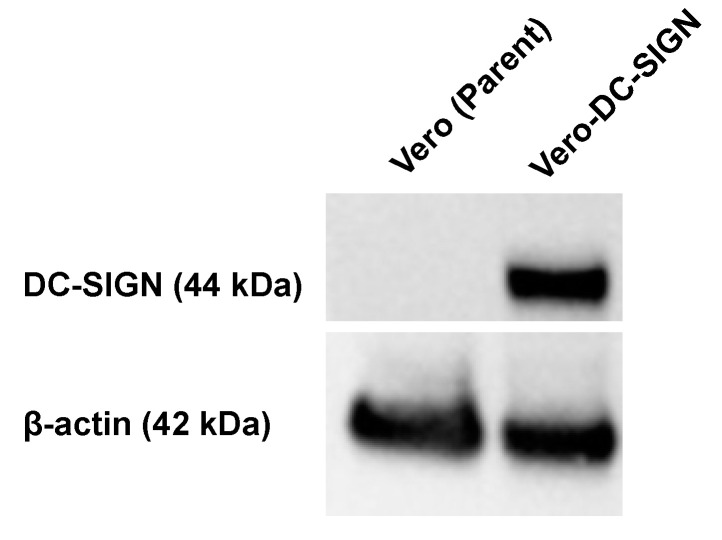
Stable expression of human DC-SIGN in Vero-DC-SIGN cells. Expression of dendritic cell-specific ICAM-3-grabbing nonintegrin (DC-SIGN) in Vero cells transfected with the human DC-SIGN genome (Vero-DC-SIGN cells) is confirmed using Western blotting. Parental Vero cells are used as negative controls. Mouse anti-DC-SIGN monoclonal antibody and mouse anti-β-actin monoclonal antibody are used as primary antibodies. A band at 44 kDa is detected for human DC-SIGN and 42 kDa for β-actin.

**Figure 2 pathogens-14-00254-f002:**
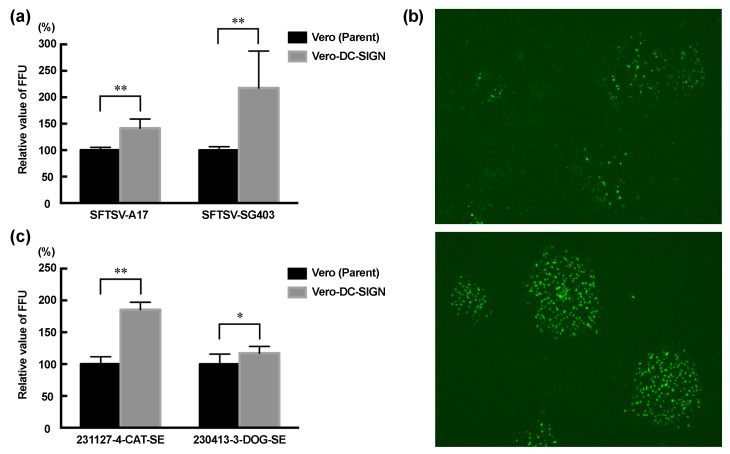
Comparison of SFTS virus infection efficiency between Vero-DC-SIGN and Vero cells. (**a**) Vero-DC-SIGN and Vero cells are infected with the same viral titers of the SFTS virus strains A17 and SG403, and the number of foci is compared between these cells; (**b**) Vero (upper panel) and Vero-DC-SIGN (lower panel) cells 4 days after inoculation with SFTS virus strain SG403 are examined under a fluorescence microscope; (**c**) equal amounts of serum from a cat with SFTS (231127-4-CAT-SE) and a dog with SFTS (230413-3-DOG-SE) are inoculated into Vero-DC-SIGN and Vero cells. Four days after the inoculation, the number of foci in these cells is compared. Bars indicate mean values and error bars indicate the standard error of mean. * indicates *p*-value ≤ 0.05; ** indicates *p*-value ≤ 0.01.

**Figure 3 pathogens-14-00254-f003:**
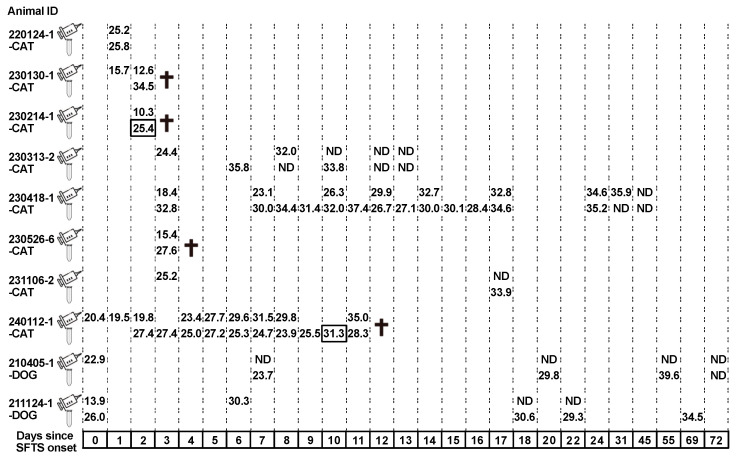
Details of 33 urine samples from which SFTS virus isolation was attempted. Days of sample collection and levels of SFTS viral RNA (Cq values) in the urine and serum of eight cats and two dogs with SFTS are depicted. The top row (syringe mark) and bottom row (spitz tube mark) indicate the Cq values of SFTS viral RNA in serum and urine, respectively. SFTS virus isolation is attempted from 33 urine samples with Cq values of 35 or less, and the infectious SFTS virus is isolated from two samples with squared Cq values (230214-1-CAT Day 2 and 240112-1-CAT Day 10). ND indicates samples for which qRT-PCR results are negative. Crosses indicate the death of the sampled animals. All animals not marked with a cross recovered from SFTS. The numbers in the boxes indicate the number of days since the onset of SFTS.

## Data Availability

The original data presented in this study are openly available in FigShare at doi.org/10.6084/m9.figshare.28435016.v1, 025.v1, 028.v1, 031.v1, and 166.v1.

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
