# Peer review of "Urine of Cats with Severe Fever with Thrombocytopenia Syndrome: A Potential Source of Infection Transmission"

_pathogens, 2025, doi:10.3390/pathogens14030254_

Round 1

Reviewer 1 Report

Comments and Suggestions for Authors

The authors report the transmission of the SFTS virus via the urine of cats. The article is novel and provides sufficient evidence. However, some results still lack convincing explanations. Our comments may help improve the manuscript to make it more understandable before acceptance.

Major comments:

1:  A definition of the “onset” is needed: is Day 0 the first day of fever and thrombocytopenia diagnosis?

2:The lack of explanation for the death of 240112-1-CAT, as it appears to be in the recovery stage based on the Cq values.

3:The time and temperature lag are not sufficient to explain why only 2 out of 33 samples are infectious to Vero-DC-SIGN cells.

4:Please provide renal function indicators, such as creatinine and blood urea nitrogen as direct evidence of urinary system damage. This would help clarify the differences between cats and dogs in terms of SFTSV progression and urine infectivity.

5:Two foci were observed in one of the four wells inoculated with eight-fold diluted urine of 240122-1-CAT-Day10, the authors should explain the possibility of false positives.

Minor comments:

1:  The missing of line numbers.

2:  The results section repeats several sentences from the methods section.

3: Such as “This study shows that SFTSV can be transmitted to humans and animals via the urine of cats infected with SFTSV.” should be revised, as the results do not yet support the conclusion.

Comments on the Quality of English Language

None

Reviewer 2 Report

Comments and Suggestions for Authors

I checked my review on the manuscript.

Comments on the Quality of English Language

Some sentence need to improve expression.
